# Bio-Fermentation Improved Rumen Fermentation and Decreased Methane Concentration of Rice Straw by Altering the Particle-Attached Microbial Community

Yao Xu [1,†], Min Aung [1,2,†], Zhanying Sun [1], Yaqi Zhou [1], Yanfen Cheng [1,3,*], Lizhuang Hao [4], Varijakshapanicker Padmakumar [5] and Weiyun Zhu [1]

1   Laboratory of Gastrointestinal Microbiology, National Center for International Research on Animal Gut Nutrition, Nanjing Agricultural University, Nanjing 210095, China; 2019105043@stu.njau.edu.cn (Y.X.); minaung.uvs@gmail.com (M.A.); sunzhanying@njau.edu.cn (Z.S.); 2019805090@stu.njau.edu.cn (Y.Z.); zhuweiyun@njau.edu.cn (W.Z.)
2   Department of Animal Nutrition, University of Veterinary Science, Nay Pyi Taw 15013, Myanmar
3   State Key Laboratory of Grassland Agro-Ecosystems, Center for Grassland Microbiome, College of Pastoral Agriculture Science and Technology, Lanzhou University, Lanzhou 730000, China
4   Key Laboratory of Plateau Grazing Animal Nutrition and Feed Science of Qinghai Province, State Key Laboratory of Plateau Ecology and Agriculture, Qinghai Plateau Yak Research Center, Qinghai 810016, China; 2009990030@qhu.edu.cn
5   International Livestock Research Institute, Nairobi 00100, Kenya; v.padmakumar@cgiar.org
*   Correspondence: yanfencheng@njau.edu.cn; Tel.: +86-25-8439-5523
†   These authors contributed equally to this work.

**Abstract:** Bio-fermentation technology has been successfully developed for ensiling rice straw; however, its effects on the particle-attached microbial community remains unknown. Therefore, rice straw (RS) and bio-fermented rice straw (BFRS) were used as substrates for in vitro rumen fermentation to investigate the effect of bio-fermentation on particle-attached microbial community, as well as their effects on gas and methane production, fermentation products, and fiber degradation. Our results have shown that total gas production, fiber degradation, and in vitro fermentation products were significantly higher ($p < 0.05$) for the BFRS than the RS, while methane concentration in total gas volume was significantly lower ($p < 0.05$) for the BFRS than RS. Linear discriminant effect size (LefSe) analysis revealed that the relative abundance of the phyla Bacteroidetes, Fibrobacteres, Proteobacteria, and Lantisphaerae, as well as the genera *Fibrobacter*, *Saccharofermentans*, and *[Eubacterium]* ruminantium groups in the tightly attached bacterial community, was significantly higher ($p < 0.05$) for the BFRS than the RS, whereas other microbial communities did not change. Thus, bio-fermentation altered the tightly attached bacterial community, thereby improving gas production, fiber degradation, and fermentation products. Furthermore, bio-fermentation reduced methane concentration in total gas volume without affecting the archaeal community.

**Keywords:** rice straw; bio-fermentation; fiber degradation; fermentation products; methane concentration; ruminal bacterial and archaea

## 1. Introduction

Ruminants are completely reliant on crops for nutrition in order to maintain proper growth, productivity, and reproduction. Rice straw is one of the most abundant crop residues in the Asian region and is used as the major roughage source for ruminant in the region. However, its recalcitrant cellulose–hemicellulose–lignin structure prevents the rumen microbes and enzymes from utilizing it, resulting in low rumen fermentation efficiency and fiber degradation [1]. Various physical and chemical approaches have been successfully developed to promote the utilization of rice straw as ruminant feed [2]; however, they have limitations. Their use leads to environmental pollution and they are high

energy demanding. Biological approaches are currently recognized as low-energy demanding and environmentally friendly, as they do not produce any effluent or fermentation inhibitors during the process [3]. Bio-fermentation is a method that involves a combination of a number of beneficial strains into a multi-strain complex and then its inoculation into a substrate, such as RS, to promote the release of cellulose and hemicellulose for use by rumen microorganisms and improve rumen fermentability by degrading lignin, thereby increasing the nutritional value of RS [4]. Xing [5] stated that inoculation with lactic acid bacterium (LAB) could decrease fiber concentration and increase dry matter and fiber digestibility of the treated rice straw (silage). Additionally, various species of *Lactobacillus* inoculants improved the silage quality and feed use efficiency of low-quality roughages (such as rice straw) [6,7].

Rumen microbes degrade the dietary carbohydrates and convert them into volatile fatty acids (VFA), ammonia nitrogen ($NH_3$-N), and microbial crude protein (MCP) for supplying nutrients to ruminants [8,9]. Rumen microbes interacting with feed particles can be functionally described as three subpopulations; those associated with rumen fluid, those loosely attached to feed particles, and those tightly attached to feed particles. Fiber-degrading enzyme activities are higher in feed particles than in the rumen fluid, thus particle-attached microbes are responsible for most of the ruminal feed digestion [10]. Without the aid of such microbes, the host ruminants cannot digest and convert plant lignocellulosic biomasses into energy and other essential metabolites [11,12]. The community composition of particle-attached microbes differs between different feeds and is likely influenced by feed chemical compositions [13,14]. Dietary composition is a major factor in determining the rumen microbiota community structure and metabolic activity [15]. Although the community composition of ruminal microbes attached to rice straw has been reported in our previous study [14], the effect of bio-fermentation on the community composition of ruminal microbes attached to rice straw remains unknown, despite the fact that the bio-fermentation method for ensiling rice straw has been successfully developed, as mentioned above.

Changes in the microbial community could contribute to understanding how foraging and ruminal microbes interact [16]. Research on the ruminal microbial community has become more convenient than before due to the advancement of microbial molecular techniques, particularly the high throughput sequencing technology. In vitro techniques are widely used to investigate digestibility of various forages and their effects on ruminal fermentation [17,18]. Therefore, the objective and the novelty of this study were to investigate the particle-attached microbial community in the bio-fermented rice straw, as well as their effects on gas and methane production, in vitro fermentation products, and fiber degradation. This information is of great significance for a deeper understanding of the colonization pattern and compositional changes of the rumen microbiome, and for the further development of bio-fermentation technology in the exploitation and utilization of roughage.

## 2. Materials and Methods

### 2.1. Substrates Preparation

The rice straw (RS) and the bio-fermented rice straw (BFRS) used in this experiment were purchased from Zhongxin Agricultural Service Professional Cooperative, Yancheng City, Jiangsu Province, China. For bio-fermenting rice straw, after harvesting rice in the field, the rice straw was picked up and shipped to the factory. The "S102 straw micro-storage" silage inoculant was supplied by the Jiangsu Academy of Agricultural Sciences. The rate of application of inoculant per kilogram of rice straw was $2 \times 10^8$ CFU/kg. Thereafter, it was wrapped in a polyethylene sheet and fermented for 40 days. The chemical compositions of the RS and the BFRS are presented in Table S1.

### 2.2. In Vitro Rumen Fermentation and Sample Collection

Four ruminally cannulated Hu sheep were used as the donor of rumen fluid. The sheep were fed a total mixed ration (40% alfalfa hay and 60% concentrate) twice a day (8:00 a.m. and 4:00 p.m.). All animal procedures were carried out using the protocols approved by the Animal Care and Use Committee of Nanjing Agricultural University, 1999. Rumen fluid was collected before morning feeding, filtered through 4 layers of sterile cheesecloth, and mixed in equal amounts from each animal. The buffered rumen inoculum was prepared by mixing the composite rumen fluid with artificial saliva (1:9, $v/v$) [19]. One gram of substrate, i.e., the RS or the BFRS, was introduced in 4 replicates into the 180 mL serum bottle, followed by the addition of 100 mL of buffered rumen inoculum under $CO_2$ steam. A blank incubation without substrate was also taken for correction of gases. All bottles were incubated at 39 °C for 24 h without shaking, and cumulative gas production was measured with a pressure transducer at 2, 4, 6, 8, 10, 12, 16, 20, and 24 h [20]. At each time point, about 5 mL gas from each bottle was collected into a gas-tight aluminum bag (Dalian Delin Gaspacking limited company, Dalian, China) for methane analysis. At 24 h of fermentation, four bottles from each group were taken out and placed in ice to cease the fermentation. From each bottle, an aliquot of about 2 mL supernatant was transferred into four 3 mL capacity tubes and stored at −20 °C for the analysis of fermentation products. Isolation of the loosely attached fraction (L) and tightly attached fraction (T) was followed by the procedure described by Larue et al. [21]. One gram of residues from each bottle was sampled into 5 mL capacity tubes before adding 5 mL phosphate buffer saline (PBS) and shaking for 30 s. Then, the supernatant obtained by centrifugation (15 min, $350 \times g$) was defined as a loosely attached fraction, while the solid residue was defined as a tightly attached fraction. These fractions were stored at −80 °C until analysis of the bacterial and archaeal community was loosely or tightly attached to the substrates. After taking the samples, the pH was immediately measured. The remaining residues were washed and dried at 105 °C until the weight remained constant, to determine the in vitro nutrient digestibility.

### 2.3. Analysis of In Vitro Fiber Digestibility of Substrates

The substrates were transferred into the 50 mL pre-weighed centrifuge tubes and dried at 105 °C. The dry matter (DM) content was analyzed according to AOAC [22], and the contents of neutral detergent fiber (NDF), acid detergent fiber (ADF), acid detergent lignin (ADL), hemicellulose (H), and cellulose (C) were analyzed by the ANKOM filter bag technique using an ANKOM 200i fiber analyzer (ANKOM Technologies, Inc., Fairport, NY, USA) [23]. Dry matter and nutrient digestibilities were calculated using the model: digestibility (%) = [(A − B)/A] × 100, where "A" meant the amount of nutrient before fermentation and "B" referred to the amount of nutrient after fermentation.

### 2.4. Analysis of Substrate Fermentation Products

The pH was measured with a pH meter (Ecoscan pH 5, Singapore). Lactate was determined with the Lactate Assay Kit (Nanjing Jiancheng Bioengineering Institute, Nanjing, China) and $NH_3$-N was measured with the method of Weatherburn [24]. Microbial crude protein (MCP) was determined with Bradford Protein Assay Kit (Beijing Solarbio Science and Technology Company, Beijing, China), while the volatile fatty acids (VFAs) were analyzed with the GC (Daojin GC2014AFsc instrument, Shimadzu, Japan). The conditions used for the GC analysis were as follows: a column temperature of 135 °C, an injection temperature of 200 °C, FID detector temperature of 200 °C, and a carrier gas ($N_2$) pressure of 0.06 MPa [25]. Methane was analyzed by the GC-TCD (Agilent 7890B, Agilent, Santa Clara, CA, USA). The condition used for GC-TCD analysis were: a column temperature of 80 °C, an injection temperature of 200 °C, a TCD detector temperature of 200 °C, and a carrier gas ($N_2$) pressure of 0.05 MPa [26].

*2.5. Analysis of Bacteria and Archaea Loosely or Tightly Attached to Substrates by Illumina Hiseq Sequencing of 16S Rrna Gene*

The samples from L (1 mL) and T (0.3 g) fractions were used for DNA extraction using the bead-beating and phenol–chloroform extraction method proposed by Zoetendal et al. [27]. After DNA extraction, a PCR thermal cycler (Eppendorf AG 22331 Hamburg, German) was used to amplify the total bacterial and archaeal 16S rRNA gene. The universal primers, 515F 5′-GTGCCAGCMGCCGCGGTAA-3′ and 806R 5′-GGACTACHVGGGTWTCTAAT-3′ [28], and Ar915F 5′-AGGAATTGGCGGGGGGAGCAC-3′ and Ar1386R 5′-GCGGTGTGTGCAAGGAGC-3′ [29], targeting the 16S rRNA gene were used to obtain the PCR amplicons of total bacteria and archaea, respectively. The PCR amplicons were then purified by means of Agencourt AMPure XP beads (Beckman Coulter, Milan, Italy). The RNA concentration was quantified with a Small RNA Kit (Agilent Technologies, 5067-1548, Beijing, China) and 2100 Bioanalyzer. Amplified libraries were sequenced on an Illumina Hiseq platform at BGI Life Tech Co., Ltd. (Beijing, China).

To remove ambiguous and low-quality sequences, the raw sequencing data were preprocessed with cutadapt v2.6 software [30]. After trimming, the sequence data were further quality-filtered to abandon reads with ambiguous, homologous sequences. If the window average quality value was <20, the end of the read sequence was truncated from the window, and the reads with a final read length <75% of the original read length were removed. Then, the reads with chimera were detected and removed by QIIME 2 software [31]. After the pretreatment described above, clean reads were grouped into amplicon sequence variant (ASV) using Vsearch software at a 99% similarity level. The representative read of each ASV was selected by using the QIIME package in bacterial and archaeal community was annotated by the SILVA 16S rRNA database and the RIM database, respectively. Alpha diversity, as indicated by the number of ASV, Evenness, Faith's phylogenetic diversity (Faith_pd), and Shannon, was calculated with QIIME 2 software. Evenness described the relative abundance of the different species making up the richness. Faith_pd was used to calculate the alpha diversity. The Shannon index was used for microbial diversity analysis. A Venn diagram was used to visualize the number of common and unique features. The unweighted UniFrac distance was used for principal coordinate analysis (PCoA) to compare the microbial communities between two groups. Linear discriminant analysis effect size (LEfSe) analysis was also employed to determine the significant differences in the bacterial community between the two groups.

*2.6. Statistical Analysis*

The data of total gas production, methane concentration, fiber degradation, fermentation products, and alpha diversity of bacteria or archaea were statistically analyzed by SPSS 25.0 (SPSS, Chicago, IL, USA) with the independent *t*-test. Statistical significance was defined at $p < 0.05$. Venn diagrams and PCoA analysis were completed by the online data visualization and analysis tool (https://www.bioincloud.tech/task-meta/ (accessed on 10 January 2022)). LEfSe analysis was performed by the online LEfSe analysis tool (http://huttenhower.sph.harvard.edu/galaxy/ (accessed on 10 January 2022)) [32]. All the data were shown as mean ± standard error of mean (SEM) and plotted in GraphPad Prism 8.0.

## 3. Results

*3.1. Effect of Bio-Fermentation on Gas Production, Methane Concentration, Fiber Degradation and In Vitro Fermentation Products of Rice Straw*

The total gas production curves (Figure 1A) showed that the values were significantly higher ($p < 0.05$) for the BFRS than the RS at the all-time points. Conversely, the methane concentration (in total gas) curves (Figure 1B) demonstrated that the values were significantly lower ($p < 0.05$) for the BFRS than the RS.

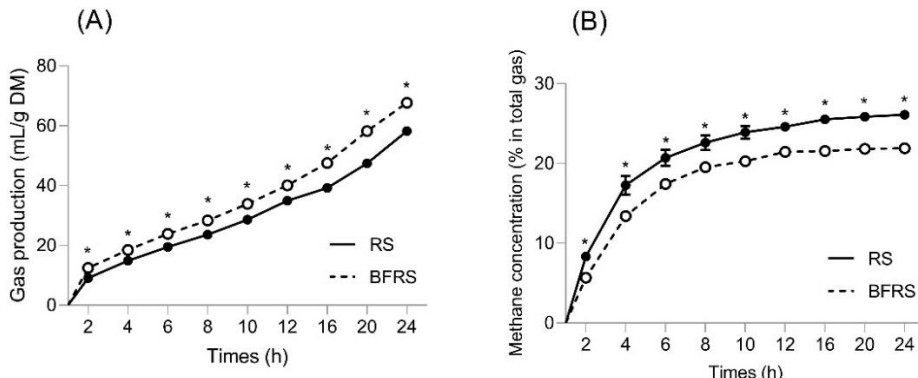

**Figure 1.** Curves of cumulative total gas production (**A**) and methane concentration (**B**) for the RS and the BFRS. Asterisks (*) indicate the significant difference ($p < 0.05$) between groups at the same time of incubation. RS: rice straw, BFRS: bio-fermented rice straw.

The digestibility of dry matter, NDF, ADF, hemicellulose, and cellulose of the BFRS were significantly higher ($p < 0.05$) than those of RS, while digestibility of ADL was not different ($p > 0.05$) between the RS and the BFRS (Table 1). The pH did not differ ($p > 0.05$) between the two groups. The fermentation products, total VFA, acetate, propionate and valerate concentrations for the BFRS were greater ($p < 0.05$) than those for the RS. However, acetate to propionate ratio, isobutyrate, butyrate, and isovalerate concentrations were not different ($p > 0.05$) between the groups. Likewise, the concentrations of lactate, $NH_3$-N, and MCP were higher ($p < 0.05$) for the BFRS than the RS (Table 1).

**Table 1.** Effect of the bio-fermentation on fiber digestibility and fermentation products in the rice straw fermentation at 24 h incubation.

| Items | Groups | |
|---|---|---|
| | RS | BFRS |
| Fiber digestibility | | |
| DMD, % | 26.44 ± 0.64 b | 31.27 ± 1.11 a |
| NDFD, % | 32.93 ± 0.86 b | 38.50 ± 1.31 a |
| ADFD, % | 23.94 ± 0.98 b | 31.76 ± 2.10 a |
| ADLD, % | 50.26 ± 1.95 | 47.06 ± 2.28 |
| HD, % | 41.65 ± 0.83 b | 45.81 ± 1.00 a |
| CD, % | 23.66 ± 1.40 b | 32.81 ± 1.03 a |
| Fermentation products | | |
| pH | 6.76 ± 0.03 | 6.75 ± 0.02 |
| Total VFA (mmol/L) | 51.78 ± 0.70 b | 56.16 ± 0.27 a |
| Acetate (mmol/L) | 30.65 ± 0.50 b | 33.52 ± 0.43 a |
| Propionate (mmol/L) | 11.65 ± 0.18 b | 13.03 ± 0.19 a |
| Acetate: Propionate | 2.63 ± 0.04 | 2.57 ± 0.04 |
| Isobutyrate (mmol/L) | 0.99 ± 0.01 | 0.97 ± 0.02 |
| Butyrate (mmol/L) | 6.52 ± 0.16 | 6.57 ± 0.19 |
| Isovalerate (mmol/L) | 1.50 ± 0.04 | 1.47 ± 0.08 |
| Valerate (mmol/L) | 0.48 ± 0.01 b | 0.60 ± 0.01 a |
| Lactate (mmol/L) | 0.16 ± 0.01 b | 0.21 ± 0.01 a |
| $NH_3$-N (mg/dL) | 9.81 ± 0.16 b | 11.02 ± 0.26 a |
| MCP (mg/dL) | 10.36 ± 0.37 b | 12.05 ± 0.31 a |

DMD: dry matter digestibility, NDFD: neutral detergent fiber digestibility, ADFD: acid detergent fiber digestibility, ADLD: acid detergent lignin digestibility, HD: hemicellulose digestibility, CD: cellulose digestibility, VFA: volatile fatty acid, $NH_3$-N: ammonia nitrogen, MCP: microbial crude protein, RS: rice straw, BFRS: bio-fermented rice straw. Different letters (a, b) indicate a significant difference at $p < 0.05$.

### 3.2. Effect of Bio-Fermentation on Loosely Attached Bacterial and Archaeal Community during In Vitro Rumen Fermentation

For the bacterial and archaeal community analyses of the loosely attached fraction, a total of 593,508 and 375,594 sequences remained from eight samples after quality filtering with a mean of 74,189 ± 21 and 44,699 ± 1577 sequences per sample, respectively. A total of 614 and 165 amplicon sequence variants (ASV) were identified at the 99% similarity level for the bacterial and archaeal communities in the loosely bound fraction, respectively. The bio-fermentation had no effect ($p > 0.05$) on the alpha diversity of bacteria and archaea loosely attached to RS (Table 2).

**Table 2.** Effect of the bio-fermentation on alpha diversity of loosely and tightly attached bacteria and archaea in the rice straw fermentation.

| Items | Loosely Attached Fraction | | Tightly Attached Fraction | |
|---|---|---|---|---|
| | RS | BFRS | RS | BFRS |
| Bacteria | | | | |
| Number of ASV | 355.75 ± 2.06 | 350.50 ± 20.34 | 448.75 ± 8.59 | 461.50 ± 3.59 |
| Evenness | 0.74 ± 0.00 | 0.72 ± 0.03 | 0.75 ± 0.00 b | 0.76 ± 0.00 a |
| Faith_pd | 27.10 ± 0.22 | 25.78 ± 0.85 | 29.99 ± 0.35 | 30.56 ± 0.18 |
| Shannon | 6.25 ± 0.03 | 6.08 ± 0.31 | 6.55 ± 0.04 b | 6.72 ± 0.04 a |
| Archaea | | | | |
| Number of ASV | 55.00 ± 12.08 | 49.00 ± 8.69 | 96.50 ± 16.66 | 96.25 ± 16.75 |
| Evenness | 0.60 ± 0.02 | 0.51 ± 0.07 | 0.58 ± 0.01 | 0.58 ± 0.01 |
| Faith_pd | 24.96 ± 6.69 | 22.98 ± 5.25 | 42.93 ± 10.16 | 45.55 ± 8.85 |
| Shannon | 3.43 ± 0.26 | 2.78 ± 0.36 | 3.81 ± 0.18 | 3.79 ± 0.11 |

ASV: amplicon sequence variant, RS: rice straw, BFRS: bio-fermented rice straw. Different letters (a, b) indicate a significant difference at $p < 0.05$.

Venn diagrams for the loosely attached fraction demonstrated that there were 427 common taxa and 57 and 130 unique taxa for the bacterial community (Figure 2A), as well as 68 common taxa and 47 and 50 unique taxa for the archaeal community (Figure 2C) of RS and BFRS, respectively. The PCoA showed that the loosely attached bacteria (Figure 2B) or archaea (Figure 2D) in the RS and the BFRS fermentations did not cluster separately.

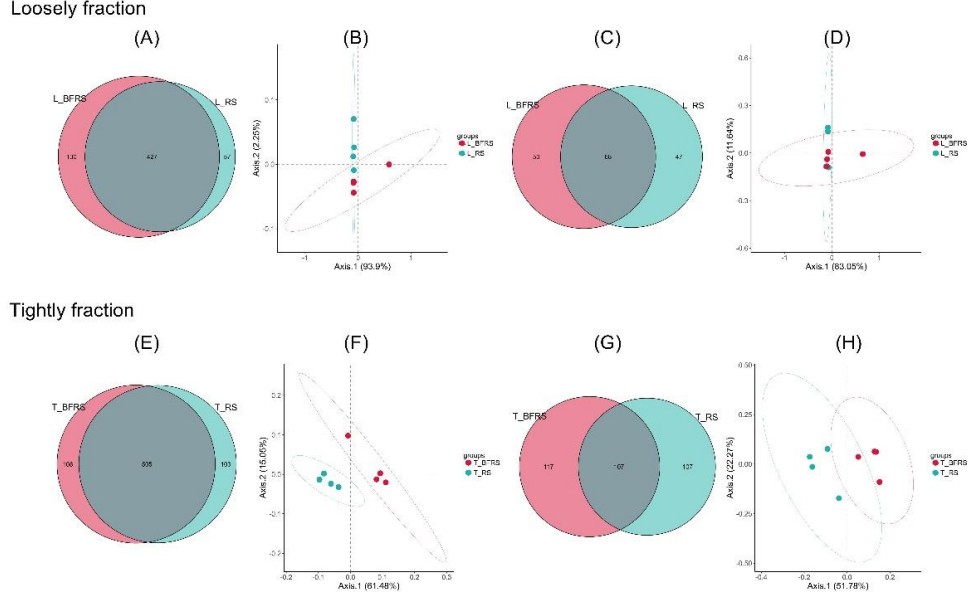

**Figure 2.** Venn diagrams of loosely attached bacteria (**A**) and archaea (**C**), and tightly attached bacteria (**E**) and archaea (**G**), and PCoA plots of loosely attached bacteria (**B**) and archaea (**D**), and tightly attached bacteria (**F**) and archaea (**H**), for the RS and the BFRS. RS: rice straw, BFRS: bio-fermented rice straw.

Six bacteria phyla were identified with relative abundances of more than 0.5% in at least one group, whereas Bacteroidetes, Firmicutes, and Proteobacteria were most abundant, accounting for 96% of total loosely attached bacteria (Figure 3A). The relative abundances of 14 bacteria genera were also above 1.0% in at least one group, whereas *Prevotella* 1, the *Rikenellaceae* RC9 gut group, and *Succinivibrio* were most abundant, representing about 62% of total loosely attached bacteria (Figure 3B). Three archaeal families were detected in this study, whereas *Methanobacteriaceae* was the most dominant family of archaea, accounting for 83% of the total loosely attached archaea (Figure 3C). Six genera of archaea were identified with relative abundance of more than 1.0% in at least one group, whereas *Methanobrevibacter*, *Methanotorris*, and *Methanomassiliicoccaceae*_Group9 were most abundant, representing about 97% of the total loosely attached archaea (Figure 3D).

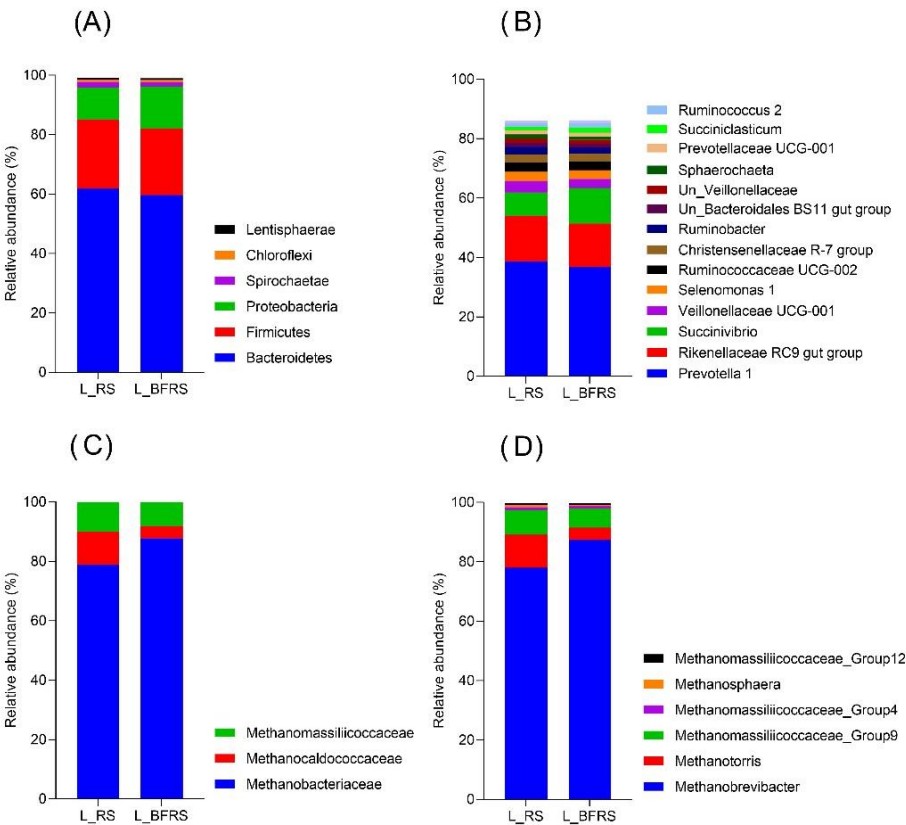

**Figure 3.** Relative abundance of loosely attached bacteria at phylum level (**A**) and genus level (**B**), and archaea at family level (**C**) and genus level (**D**) for the RS and the BFRS (the average relative abundances of phyla or family >0.5%, and genera >1.0% in at least one group are presented). RS: rice straw, BFRS: bio-fermented rice straw.

The LEfSe analysis (from phyla to genus level) was performed for loosely attached bacterial and archaeal communities. Only uncultured *Lachnospiraceae* belonging to the phyla Firmicutes for the bacterial community (Figure S1A) and *Methanosphaera* belonging to the family *Methanocaldococcaceae* for the archaeal community (Figure S1B) were higher in RS.

### 3.3. Effect of Bio-Fermentation on the Tightly Attached Bacterial and Archaeal Community during the In Vitro Rumen Fermentation

For the bacterial and archaeal community analyses of the tightly attached fraction, a total of 593,631 and 347,606 sequences remained from eight samples after quality filtering with a mean of $74,204 \pm 16$ and $43,451 \pm 631$ sequences per sample, respectively. Totals of 716 and 331 amplicon sequence variants (ASV) were identified at the 99% similarity level for the bacterial and archaeal communities of the tightly attached fraction, respectively.

Alpha diversity of bacteria tightly attached to the RS was significantly changed by the bio-fermentation, whereas Evenness and Shannon were significantly higher ($p < 0.05$) for the BFRS than the RS. However, alpha diversity of archaea tightly attached to RS did not differ ($p > 0.05$).

Venn diagrams for tightly fraction revealed that there were 505 common taxa and 103 and 108 unique taxa for bacteria community (Figure 2E), and 107 common taxa and 107 and 117 unique taxa for archaea community (Figure 2G) in the RS and BFRS fermentations, respectively. The PCoA displayed that tightly attached bacterial communities of the RS and the BFRS fermentations were clustered into two separate groups (Figure 2F), while archaea were not clustered separately (Figure 2H).

Eight bacteria phyla were identified with relative abundances of more than 0.5% in at least one group, whereas Bacteroidetes and Firmicutes were most abundant, accounting for 89% of total tightly attached bacteria (Figure 4A). The relative abundances of 17 bacteria genera were also above 1.0% in at least one group, whereas *Prevotella* 1 was the most abundant accounting for 29%, followed by the *Rikenellaceae* RC9 gut group (14%), *Probable* genus 10 (8%), the *uncultured Bacteroidales* S24-7 group (8%), and the *Christensenellaceae* R-7 group (5%) of the total tightly attached bacteria (Figure 4B). Three archaeal families were detected in this study, whereas *Methanobacteriaceae* was the most dominant family of archaea, accounting for 70% of the total tightly attached archaea (Figure 4C). Seven genera of archaea were identified with relative abundance of more than 1.0% in at least one group, whereas *Methanobrevibacter* and *Methanotorris* were the most abundant, representing about 88% of the total tightly attached archaea (Figure 4D).

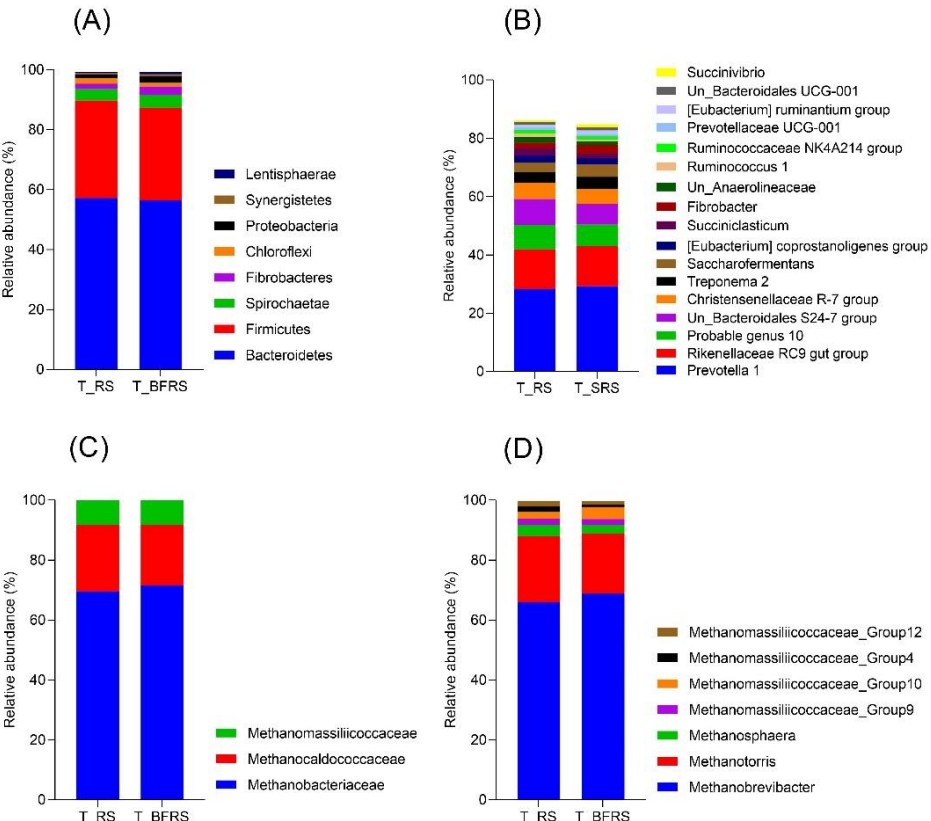

**Figure 4.** Relative abundance of tightly attached bacteria at phylum level (**A**) and genus level (**B**), and archaea at family level (**C**) and genus level (**D**) for the RS and the BFRS (the average relative abundances of phyla or family > 0.5%, and genera > 1.0% in at least one group are presented). RS: rice straw, BFRS: bio-fermented rice straw.

The LEfSe analysis (from phyla to genus level) was performed for tightly attached bacterial and archaeal community. For the bacterial community, eight bacterial phyla

were significantly different, whereas two phyla were higher for the RS and six phyla were higher for the BFRS. Among those bacterial phyla with the relative abundance of >0.5% in at least one sample, Chloroflexi was higher for the RS, and Bacteroidetes, Fibrobacteres, Proteobacteria, and Lentisphaerae were higher for the BFRS (Figure 5A). At the genus level, 44 genera were significantly different, whereas 21 genera were higher for the RS and 23 genera were higher for the BFRS (Figure 5A). Among those bacterial genera with the relative abundance >1.0% in at least one sample, *Ruminococcus* 1 and *Prevotellaceae UCG-001* were higher for the RS, and *Fibrobacter*, *Saccharofermentans*, and *[Eubacterium]* ruminantium groups were higher for the BFRS (Figure 5B). For the archaeal community, the genera group 4 and group 10, belonging to the *Methanomassiliicoccaceae* family, were higher for the RS and the BFRS, respectively (Figure S1C).

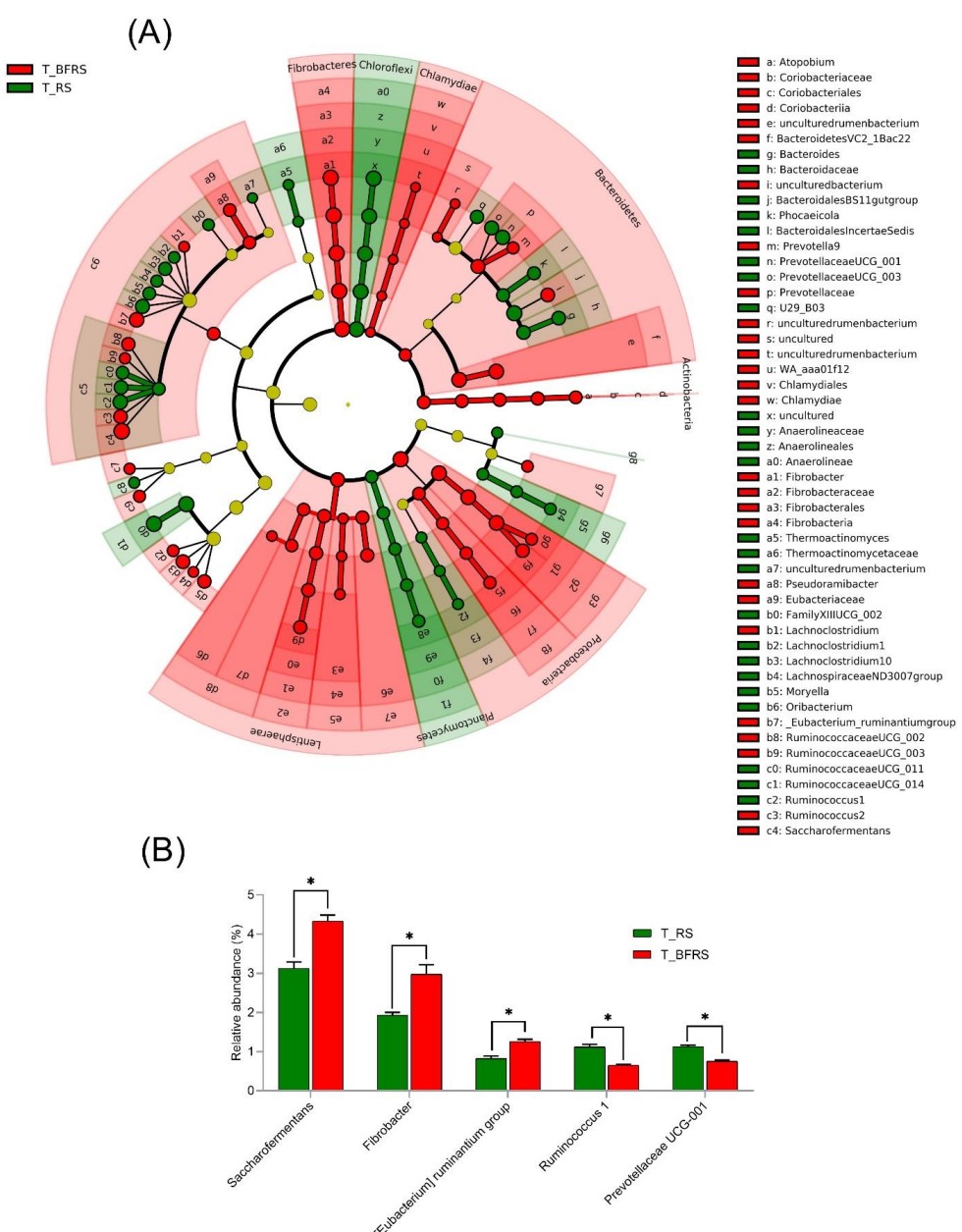

**Figure 5.** LEfSe analysis of the tightly attached bacterial community in the RS and the BFRS fermentations. Cladogram showing significantly enriched bacterial taxa from phylum to genus level (**A**), and comparison of significantly different bacterial genera (**B**). Asterisks (*) indicate the significant difference ($p < 0.05$) between groups. RS: rice straw, BFRS: bio-fermented rice straw.

## 4. Discussion

The in vitro rumen fermentation technique does not necessitate expensive equipment, and the large numbers of samples can be incubated and analyzed at the same time. This technique simulates the rumen fermentation process and has been used to evaluate the potential of feed for supplying nutrients to ruminants. Likewise, this technique can be used to investigate feed methane production, which is a waste of feed energy and a source of global warming. Thus, in this study, the in vitro rumen fermentation technique was used to evaluate the total gas and methane production, fiber degradation, and microbial community attached to bio-fermented rice straw.

The chemical composition of substrates, particularly the fiber content of feed, was found to be negatively associated with the gas production [33,34]. The low content of fiber can facilitate the utilization of feed by ruminal microbes, resulting in higher fermentation rates [35]. This assertion supported our results on gas production, which was higher for BFRS, with lower NDF and hemicellulose, and higher NDS contents. Furthermore, the gas production is a reflection of carbohydrate fractions and degradation [36,37]. Subsequently, the lignocellulose degradation of substrates in rumen in vitro fermentation was investigated. As the results show, a higher degradation of DM, NDF, ADF, hemicellulose, and cellulose was observed for the BFRS compared to the RS. This is likely due to the higher NDS and lower fiber (NDF, ADL, and hemicellulose) contents of the BFRS, which could lead to increased DM degradation, gas production, and fermentation rate. Zhang et al. [38] reported that RS with higher NDS content had higher in vitro DM degradation and gas production. Moreover, Gharechahi et al. [39] stated that the total DM degradation was mainly determined by NDF content of forages, which is consistent with lower NDF content and higher DM degradation of BRFS observed in our study.

The fermentation products are negatively related to the fiber contents [40], and positively correlated with non-fibrous carbohydrate (NFC) content [41] of substrates. Lactate, an intermediary product, is mainly produced from the fermentation of easily fermentable feed, and then metabolized to VFAs, the end products of rumen microbial fermentation [23]. This is congruent with our findings, which showed that the BFRS with lower NDF and higher NDS contents had higher lactate and VFA, especially total VFA, acetate, and propionate concentrations, than those for the RS. Protease hydrolyzes proteins into amino acids and peptides, while microbial deamination converts some of the amino acids to ammonia [42]. A portion of ammonia is converted to MCP by bacteria, while the rest is absorbed into the blood and contributes to the rumen nitrogen cycle [43]. In our study, protein contents of two substrates were similar; however, $NH_3$-N and MCP were higher in the BFRS than the RS fermentations. This could be attributed to the enhanced protein breakdown into ammonia by the bio-fermentation and higher MCP synthesis from ammonia.

The rumen microbes degrade the dietary nutrients and convert them into VFA, $NH_3$-N, and MCP for supplying nutrients to ruminants. Carbohydrates, especially fibrous carbohydrates, are mainly degraded by the microbes attached to feed particles [8,9]. Among the alpha diversity of bacteria and archaea in both the loosely and tightly attached fractions, the bacterial alpha diversity (Evenness and Shannon) in the tightly attached fraction was significantly higher for the BFRS than the RS, which was supported by the PCoA plots, where the tightly attached bacterial communities of the RS and the BFRS were clustered into two separate groups.

The bacterial compositional analysis showed that Bacteroidetes and Firmicutes were the most abundant bacterial phyla in both the loosely and tightly attached fractions. This finding is consistent with the report of Cheng et al. [14], who found Bacteroidetes and Firmicutes to be the most dominant bacterial phyla present both in loosely and tightly attached fractions to rice straw. Bacteria belonging to these two phyla, Bacteroidetes and Firmicutes, were associated with the fiber degradation and polysaccharide degradation [44,45], and are considered to be the primary degrader of complex soluble polysaccharides in plant cell walls [46,47]. At the genus level, the *Prevotella* 1 and *Rikenellaceae* RC9 gut groups

were also the most abundant bacterial genera in both loosely and tightly attached fractions. Xie et al. [48] reported that the *Prevotella* 1 and *Rikenellaceae* RC9 gut groups were most abundant bacterial genera in the rumen of Hu sheep. *Prevotella* has a great functional versatility and is mainly involved in carbohydrate and nitrogen metabolisms in the rumen, as well as in producing a variety of enzymes involved in the degradation of starch, proteins, peptides, and hemicellulose [49,50]. Propionate synthesis by *Prevotella* spp. is important for maintaining glucose homeostasis in host animals through gluconeogenesis [49]. The *Rikenellaceae* RC9 gut group is associated with primary or secondary carbohydrate degradation and protein fermentation [51,52].

The LEfSe analysis of loosely attached bacterial community demonstrated that only taxa, uncultured *Lachnospiraceae* belonging to the phyla Firmicutes, was higher in the RS. In the tightly attached bacterial community, the bio-fermentation improved the relative abundance of the phyla Bacteroidetes, Fibrobacteres, Proteobacteria, and Lantisphaerae, as well as the genera *Fibrobacter*, *Saccharofermentans*, and *[Eubacterium]* ruminantium groups for rice straw. This is likely due to the destruction of lignocellulosic structure of rice straw by the bio-fermentation treatment, which facilitated the colonization and growth of microorganisms [53]. Bacteroidetes have high saccharolytic activities and are involved in pectin, hemicellulose, and cellulose degradation [54,55], which may be attribute to the exposure of lignocellulosic structures in BFRS. Proteobacteria is mainly involved in the nitrogen metabolism in the rumen [56], and higher concentration of $NH_3$-N in BFRS may be responsible for the increase in Proteobacteria. Fibrobacteres and its genus, *Fibrobacter*, are recognized as the cellulolytic bacteria, which produces cellulolytic enzymes capable of degrading cellulose [57,58], and converts feeds into VFA [59]. This also explains why the BFRS group possesses higher DMD, CD, HD, and VFAs. *Saccharofermentans* are the sugar-fermenting bacteria and utilize glucose as a substrate for fermentation to produce acetate [60]. Moreover, they are associated with the digestibilities of ADF and cellulose [61], which is consistent with the higher digestibilities of ADF and cellulose for the BFRS observed in our study. *[Eubacterium]* ruminantium, the hemicellulose fermenter, is recognized as the fibrolytic bacteria [62]. Its higher relative abundance in BFRS indicates its involvement in hemicellulose degradation. As mentioned above, bio-fermentation increased the relative abundances of bacteria related to cellulose and hemicellulose degradation, as well as nitrogen metabolism, thereby improving the fiber degradation and fermentation profile.

Feed fermentation, degradation, and digestion depend on the attachment of rumen bacteria to feed particles. The makeup of the rumen microbial community, which influences fermentation products and ruminal pH, could be altered by the physicochemical qualities of feed [15,63]. Changes in the microbial community could help in understanding how forage and ruminal microorganisms interact [16]. However, the effect of bio-fermentation on changes in physicochemical properties of rice straw was not studied in this work, which needs investigation for a better understanding of the bio-fermentation process that increased the nutritional value of rice straw.

Methane emission from enteric fermentation is the waste of energy ingested by ruminant, and is also a significant source of greenhouse gas, accounting for approximately 10–12% of global methane emissions [64]. In our study, the methane concentration in total gas volume was lower for the BFRS than the RS, without changes in the archaeal composition. This is likely due to the higher concentration of organic acids in the BFRS fermentation, which could inhibit the ability of archaea to use $H_2$ for methane production, without affecting the archaeal population [65].

As for the archaeal composition, only the three families were detected in both loosely and tightly attached fractions, where *Methanobacteriaceae* was the most abundant, followed by *Methanocaldococcaceae* and *Methanomassiliicoccaceae*. At the genus level, *Methanobrevibacter* and *Methanotorris* were the most abundant archaeal genera in both loosely and tightly attached fractions. Huang and Li [66] stated that *Methanobrevibacter* is the most dominant archaeal genus in the rumen of Tibetan sheep and Gansu Alpine Finewool sheep. Xie et al. [48] also reported that the *Methanobacteriales* and its species, *Methanobrevibacter*

*gottschalkii* clade, *Methanobrevibacter boviskoreani* clade, and *Methanobrevibacter ruminantium* clade, were most abundant archaeal taxa in the rumen of Hu sheep. *Methanobrevibacter* belonging to the family *Methanobacteriaceae,* mainly utilizes hydrogen and carbon dioxide to produce methane [67,68]. *Methanotorris* belonging to the family *Methanocaldococcaceae* is the methanogen that uses hydrogen, carbon dioxide, and formate as the substrates to produce methane [69]. These two archaeal taxa are recognized as the methanogens with high methane conversion capacity [70]. According to the LEfSe analysis of the loosely attached fraction, the bio-fermentation reduced the relative abundance of *Methanosphaera* belonging to the family *Methanocaldococcaceae*, which has the greater methane conversion capacity as mentioned above. In the tightly attached fraction, the genera group 4 and group 10, belonging to the family *Methanomassiliicoccaceae*, were higher for the RS and the BFRS, respectively. These archaeal taxa produce methane by the reduction of methanol or methylamines, which contributes to lower methane emissions [71,72]. Jin et al. [25] reported that *Methanomassiliicoccales* are probably less important contributors to the rumen methane emission of Chinese goats because of their low relative abundance among the archaeal community. For this reason, the bio-fermentation could have reduced the methane concentration in the total gas volume for the RS.

## 5. Conclusions

The bio-fermentation altered the tightly attached bacterial community, thereby improving gas production, fiber degradation, and fermentation products. Furthermore, the bio-fermentation reduced methane concentration in total gas volume without affecting the archaeal community. Our results confirmed that bio-fermentation improved rumen fermentation effectively, and deepened our understanding of the colonization pattern and compositional changes of the rumen microbiome. This suggests the need to conduct further research on the effect of the bio-fermentation on the physicochemical structure of rice straw, as well as hydrogen production and utilization by ruminal microorganisms, with a focus on methane conversion efficiency.

**Supplementary Materials:** The following supporting information can be downloaded at: https://www.mdpi.com/article/10.3390/fermentation8020072/s1, Table S1. Chemical compositions of substrates. Figure S1. LEfSe analysis of the loosely attached bacteria (A), and archaea (B), and tightly attached archaea (C). RS: rice straw, BFRS: bio-fermented rice straw.

**Author Contributions:** Methodology, Y.X. and M.A.; investigation, Y.X., M.A., L.H., Z.S. and V.P.; formal analysis, Y.X., M.A. and Y.Z.; data curation, Y.X., M.A. and Y.Z.; writing—original draft preparation, Y.X.; writing—review and editing, M.A., Z.S., L.H., Y.C., V.P. and W.Z.; supervision, Y.C.; project administration, Y.C. and V.P.; funding acquisition, Y.C., W.Z. and V.P. All authors have read and agreed to the published version of the manuscript.

**Funding:** This research was funded by the National Natural Science Foundation of China, grant number 32061143034.

**Institutional Review Board Statement:** Not applicable.

**Informed Consent Statement:** Not applicable.

**Data Availability Statement:** The data of the 16S rRNA gene sequences of gut microbiota presented in this study can be available from the NCBI database (https://www.ncbi.nlm.nih.gov/genome/ (accessed on 10 January 2022)) under accession number PRJNA795694.

**Acknowledgments:** This research has been funded by the National Natural Science Foundation of China through project number 32061143034.

**Conflicts of Interest:** The authors declare no conflict of interest.

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
