# Peer review of "Bio-Fermentation Improved Rumen Fermentation and Decreased Methane Concentration of Rice Straw by Altering the Particle-Attached Microbial Community"

_fermentation, doi:10.3390/fermentation8020072_

Round 1

Reviewer 1 Report

The submitted manuscript clearly presents originality and interest for people working on the fields of the study of bacterial inoculants and ensiling raw plan biomass for biogas purposes. The main advantage of the paper is the identification of microbiota in fermented and non-fermented rice straw. The authors aimed to conlude that in the tightly attached bacteria community some genera were represented more frequently than others. Although the majority of the discussion is perfectly written, still, in my opinion the discussion lacks in depth when Authors describe differences in microbial community. I woulde recommend to discuss why did it happen that some microorganisms were present in BFRS and some not. What are the consequences of growing of particular microorganisms in fermented product?

Specific comments:

  • materials and methods section - please provide more details on procedure in reference [21];
  • when you are showing differences between groups I would recommend to use letters or an asterisk, please remove p-value from the table 1 and 2
  • table 1 - use lowercase when writing NH3
  • table 2 - there is no explanations what does mean "evenness" etc.
  • table S1 - please add units for cellulose and hemicellulose

I hope my comments will help the author improve their manuscript

Author Response

Dear Editor Kelsey Chen and Reviewers:

We are very grateful to you for reviewing the paper so carefully.

We have carefully considered the suggestion of Reviewers and make some changes.

Reviewer 2 Report

This is an interesting paper. These are my remarks and suggestions:

  • The title is too long; please give a concise one
  • In the Introduction section, mention this study's novelty.
  • Line 21. Use an impersonal form of writing
  • Line 27 LEfSe, please mention the explanation of the abbreviation on the first use
  • Line 55 The microorganism name should be italicized
  • In the Conclusion section mention how the study advances the knowledge in the field

Author Response

(The authors gave the same response as above.)
